# Camellia Seed Cake Extract Supports Hair Growth by Abrogating the Effect of Dihydrotestosterone in Cultured Human Dermal Papilla Cells

**DOI:** 10.3390/molecules27196443

**Published:** 2022-09-29

**Authors:** Ling Ma, Huchi Shen, Chengge Fang, Timson Chen, Jing Wang

**Affiliations:** 1Adolph Innovation Laboratory, Guangzhou Degu Personal Care Products Co., Ltd., Guangzhou 510000, China; 2Key Laboratory of Synthetic and Biological Colloids, Ministry of Education, School of Chemical and Material Engineering, Jiangnan University, Wuxi 214122, China

**Keywords:** androgenetic alopecia, dermal papilla cells, dihydrotestosterone, camellia seed cake, cell senescence

## Abstract

Autocrine and paracrine factors play key roles in the process of Androgenetic alopecia (AGA), which are secreted by balding dermal papilla cells (DPCs) after dihydrotestosterone (DHT) induction. Camellia seed cake is an oriental oil extraction byproduct, and its extract has been traditionally used to wash hair in China. This study elucidated the hair growth-promoting effects of Camellia seed cake extract (CSCE) in DHT-treated cultured DPCs and its underlying mechanisms. The effect of CSCE on cell viability and release of inflammatory factors IL-6 and IL-1α was performed on human dermal papilla cells (DPCs) incubated with DHT. Relative expression of bax, bcl-2, p53, androgen receptor (AR) and 5α- reductase type II (SRD5A2) was determined by PCR. Senescence-associated was examined by β-galactosidase (SA-β-Gal) assays. CSCE restored DHT-induced cell damage in a dose-dependent manner, and effectively reduced the production of IL-6 and IL-1α in DHT-treated DPCs. CSCE exhibited an anti-apoptotic effect, which increased the expression of bcl-2, and decreased the expressions of bax and p53 in DHT-incubated DPCs. CSCE also showed an anti-androgenic effect reversing the increase in AR and SRD5A2 expressions in DPCs driven by DHT incubation. In addition, CSCE inhibited the β-galactosidase enzyme activity and slowed down the cell senescence of DPCs which is crucial for AGA progression. In this study, we found that CSCE may have the potential to prevent and alleviate AGA by abrogating the effect of DHT in cultured DPCs.

## 1. Introduction

As a common, non-physiological alopecia, Androgenetic alopecia (AGA) is manifested as diffuse alopecia on the top of the head and miniaturization of hair follicles, which is often accompanied by symptoms such as dandruff and greasy hair. The symptoms of AGA could gradually get more serious with aging. AGA has a great influence on people’s appearance and even mental health [1]. In recent years, the prevalence rate has been increasing and showing a trend of younger [2]. Although the pathogenesis and underlying molecular mechanism of AGA are not fully elucidated, studies strongly suggested that abnormal androgen metabolism is inextricably associated with AGA [3].

Hair follicle growth shows a unique periodicity, and is influenced by androgens. By affecting the activities of the DPCs, androgens also indirectly control hair growth. DHT, a metabolite of testosterone (T) by 5α- reductase type II (SRD5A2) in hair follicles, is considered to be the major cause of hair loss. Recently, it has been shown that the level of DHT in patients with AGA was higher than that in normal people. It was also suggested that the DHT-driven alteration in autocrine and paracrine inhibitory factors released by DPCs may be the key to AGA [4,5]. In the nucleus, AR undergoes a series of conformational changes upon DHT binding, which leads to changes in DNA transcription, thereby affecting protein expression and hair follicle hair growth [6,7]. Autocrine and paracrine factors play key roles in the process of Androgenetic alopecia (AGA), which are secreted by balding dermal papilla cells (DPCs) after dihydrotestosterone (DHT) induction. In a study by Kwack [8], IL-6 expression was found to be upregulated in balding DPCs compared to non-balding cells. The expression of IL-6 was shown to be induced by DHT in cultured DPCs, and IL-6 reduced the activity of keratinocytes and inhibited hair shaft elongation and the proliferation of stromal cells in cultured hair follicles.

Cellular senescence, which is a potential cause of biological aging, also plays an important role in the process of AGA. DPCs from male patients with AGA showed premature aging [9,10,11]. The vast majority of normal cells have limited ability to divide, and will enter a state of cell senescence after being unable to divide. Senescent cells have a special cell cycle distribution, which cannot proliferate or divide under some conventional stimulation while the proliferation of DPCs is closely related to hair growth. Stresses such as oxidative stress and genotoxic stress can induce cellular senescence. In general, these findings suggested that DHT might affect DPCs and the healthy growth of hair.

DHT dependence of AGA has been well demonstrated [12,13], and compounds that could prevent AGA have been widely explored. Natural products provide a rich source of useful bioactive compounds which are used in traditional medicine. Natural product extracts such as Plumbago zeylanica [14] and Polygonum multiflorum [15] have the ability to alleviate the effect of DHT on DPCs. As a by-product during the oil refining process of Camellia oleifera Abel seed, Camellia seed cake extract has been traditionally used for hair wash in China for over one thousand years. Our recent work has found that Camellia seed cake extract facilitates DPCs proliferation and promotes hair growth in C57BL/6J mice [16]. However, the effects of Camellia seed cake extract (CSCE) on AGA have not been reported till now. In the present work, we demonstrated that the androgen dihydrotestosterone not only induced different degrees of damage to DPCs, stimulated cell inflammatory response, regulated the expression of molecules related to dermal papilla cell apoptosis and AGA-related signaling, but also induced DPCs senescence. All the above-mentioned effects caused by dihydrotestosterone could be effectively alleviated by CSCE.

## 2. Results

### 2.1. Characterization Analysis of CSCE

According to the UPLC-MS analysis, we found that there are main two flavonoids that may play a major role in CSCE (Figure 1a), and the corresponding mass spectra of compound **1** and compound **2** are displayed in Appendix A. The molecular ion peak *m/z* in positive and negative ion mode includes: compound **1**, 757[M + H]^+^, 755[M − H]^−^, and it can be determined that the molecular weight of compound **1** is 756; compound **2**, 727[M + H]^+^, 725[M − H]^−^, the molecular weight of compound **2** was determined to be 726. The positive ion mass spectrum of compound **1** further gives three characteristic fragment ion peaks (*m/z*), which are 595[M-Glu/Gla]^+^, 449[M-Glu/Gla-Rha]^+^, and 287[M-Glu/Gla-Rha-Glu/Gla]^+^; the flavonoid compound **1** is speculated to have kaempferol, two galactose (or glucose) and one rhamnotriglycoside structure. The fragment ion peaks (*m/z*) of the positive ion mass spectrum of compound **2** are 595[M-Xyl]^+^, 449[M-Xyl-Rha]^+^ and 287[M-Xyl-Rha-Glu/Gla]^+^, respectively, The flavonoid compound **2** was assumed to have kaempferol, one xylose, one rhamnose and one glucose (or galactose) triglycoside structure. It was initially identified as compound **1** (at Rt 1.59 min) kaempferol-3-*O*-[2-*O*-β-d-galactopyranosyl-6-*O*-α-l-rhamnopyranosyl]-β-d-glucopyranoside and compound **2** (at Rt 1.85min) kaempferol-3-*O*-[2-*O*-β-d-xylopyranosyl-6-*O*-α-l-rhamnopyranosyl]-β-d-glucopyranoside according to some previous literature [17].

### 2.2. CSCE Alleviated DHT-Induced Cell Damage

The cytotoxicity of dihydrotestosterone (DHT) was examined and the results are presented in Figure 2a. DHT did not show any cytotoxicity to DPCs at the concentration of 0.8 μg/mL (*p* < 0.05). However, when the concentration was higher than 5 μg/mL, DHT exhibited inhibition activity on cell growth. The effect of CSCE on DHT-induced cell damage was then investigated in DPCs. The cells were incubated with 10 μg/mL of DHT for 24 h before being treated with different concentrations of CSCE. The cell viability was shown in Figure 2b. It was found that the cell survival rate was only 83.64% after incubation with DHT, and the cell viability was restored by CSCE in a dose-dependent manner. The cell survival rate reached 98.51% after administration of 20 μg/mL CSCE.

### 2.3. CSCE Alleviated DHT-Induced Cell Inflammation 

To determine the effect of DHT on cell inflammation response, DPCs were treated with various concentrations of DHT, and the production of IL-6 and IL-1α inflammatory factors was evaluated. As shown in Figure 3a,b, secretions of IL-6 and IL-1α increased after treatment with 80–200 ng/mL DHT (*p* < 0.01), compared with the control group without DHT. This indicated that DHT induced inflammatory response in DPCs. Maximum secretions of IL-6 and IL-1α were observed when treated with DHT at the concentration of 200 ng/mL. Therefore, 200 ng/mL DHT was finally selected for subsequent assays.

Based on the DHT-induced cell inflammation model established above, the cells were incubated with 200 ng/mL DHT as a stimulator, and the results are shown in Figure 3c,d. As shown in Figure 3c,d, the contents of IL-6 and IL-1α were greatly promoted by DHT stimulation (*p* < 0.001), while the IL-6 secretion was significantly decreased when co-incubated with CSCE at concentrations of 5, 10 and 20 μg/mL (*p* < 0.01). The secretion of IL-1α was significantly inhibited by 10 and 20 μg/mL CSCE (*p* < 0.01) (Figure 3d).

### 2.4. CSCE Downregulated Bax and p53 mRNA While Upregulated bcl-2 mRNA

We further investigated whether genes related to cell survival or apoptosis, bcl-2, bax and p53, were involved in CSCE mediated inhibition of DHT-induced cell apoptosis. The Bcl-2 family proteins are important regulators of cell apoptosis. These proteins interact with each other to control the permeability of the inner and outer mitochondrial membranes by activating a series of downstream genes. Among the Bcl-2 family proteins, bcl-2 is an anti-apoptotic protein, which is highly expressed in dermal papilla, melanocytes and hair matrix. Bcl-2 could activate DNA repair, promote cell proliferation and inhibit apoptosis, thus facilitating hair growth [15,16]. The pro-apoptotic protein bax binds to the bcl-2 protein on the mitochondrial membrane, forming a bax-bcl-2 dimer and inhibiting the anti-apoptotic effect of bcl-2 [18,19,20]. Studies have shown that p53 is a negative regulator of the cell growth cycle, with low expression in normal cells and high expression when cells are subjected to stress stimuli such as DNA repair and apoptosis [21,22]. p53 is involved in the apoptosis of hair follicles during anagenesis [23,24,25]. As shown in Figure 4, incubation with DHT (200 ng/mL) significantly increased the expressions of p53 and bax (*p* < 0.01) while decreasing the expression of bcl-2 (*p* < 0.01), indicating that DHT induces DNA damage as well as cells apoptosis in DPCs. In contrast, upregulation of bcl-2 and downregulation of bax and p53 were detected when treated with CSCE, which suggested that CSCE inhibited DHT-induced cell apoptosis in DPCs.

### 2.5. Anti-Androgenic Effect of CSCE

In the cytoplasm of DPCs, testosterone is converted by type II 5α-reductase to a more potent form, DHT, which binds to AR and forms an androgen-receptor complex before entering the nucleus. In the nucleus, the androgen-receptor complex binds to hormone response elements at specific gene loci, and inhibits the growth of hair follicles, causing hair loss called AGA. In the present study, the anti-androgenic effects of CSCE in DPCs were investigated. The results showed that treatment with 200 ng/mL DHT upregulated AR and SRD5A2 gene expressions. However, CSCE abrogated the DHT-induced increments of SRD5A2 and AR gene expressions in DPCs (Figure 5), which suggested that CSCE may alleviate hair follicle miniaturization mediated by DHT.

### 2.6. CSCE Delayed DHT-Induced Cell Senescence

It was reported that premature senescence was observed in cultured DPCs derived from AGA patients. Stress-induced senescence is believed to have originated from programmed senescence, which is crucial for tissue remodeling during development. In our study, the implications of DHT and CSCE on cell senescence were evaluated in DPCs. The results of β-galactosidase staining are shown in Figure 6. The ratio of β-galactosidase-positive DPCs was increased by DHT in a concentration-dependent manner, indicating that DHT induced the senescence of DPCs. Then, various concentrations of CSCE were administered and incubated with the cells. Compared with the model group, the numbers of positive cells were decreased in groups treated with 5, 10, and 20 μg/mL CSCE. This revealed that DHT promoted cell senescence in DPCs, while CSCE inhibited the enzymatic activity of β-galactosidase and delayed cell senescence.

## 3. Discussion

The effects of DHT on androgen-sensitive DPCs were the main cause of hair miniaturization and hair loss in AGA. Symptoms of AGA gradually become more serious with aging, and premature senescence was observed in DPCs of AGA patients [10,26]. Androgen is a key regulator of hair growth. By inducing premature catagen and reducing anagen duration, androgen shortens the hair cycle and accelerates hair loss [6]. Our recent work demonstrated that Camellia seed cake extract (CSCE) exerts a hair growth-promoting effect in vitro and in vivo by proliferating DPCs through the ERK and AKT signaling pathways and regulating the expression of growth factors [16]. However, the effects of CSCE on AGA have not been addressed till now. 

Previous studies have shown that flavonoids from herbal plants are a promising source of compounds for AGA treatment due to their outstanding antioxidant and anti-inflammatory properties. Flavonoid glycosides extracted from Dicerocaryum senecioides leaves showed significant hair regeneration activity in BalB/c mice [27]. It was also reported that flavonoids stimulated hair growth by inhibiting the activity of 5α reductase type II [28,29]. Thus, we speculated that flavonoids in Camellia seed cake may be the reason that CSCE abrogated the multiple effects of DHT in DPCs. It is revealed that there are mainly two unique flavonoids in Camellia seed cake by UPLC-MS which possibly are the active ingredients. Then we found that CSCE has the potential to prevent and alleviate AGA by abrogating the effect of DHT in cultured DPCs. The report showed the effects of DHT on DPCs and the anti-androgenic effects of CSCE. DHT levels in AGA patients were elevated compared with that in normal people, suggesting that a high concentration of DHT was involved in the formation of AGA. DHT showed an inhibitory effect on cell growth when the concentration was higher than 5 μg/mL (Figure 2a), and the secretions of IL-6 and IL-1α inflammatory factors were increased by DHT-driven alterations (Figure 3a,b). Previous studies have found that inhibitory paracrine and autocrine factors produced by DHT-driven DPCs may be responsible for the induction of AGA [5,30]. DHT induces miniaturization of the dermal papilla and hair follicle, leading premature catagen phase [31]. Transforming growth factor-β1/β2 (TGFβ1/2), which is an inhibitor for epithelial cell growth, was found to increase androgenic DPCs [32,33], and DHT-induced Dikkopf1 led to apoptosis in keratinocytes [34]. Inhibitory autocrine factors were secreted by DPCs at sites of AGA, which in turn inhibited the growth of DPCs [35]. Through ELISA screening assays, Kwack et al. [8] found that secretion of IL-6 was higher in DPCs at sites of AGA than that in normal DPCs. IL-6 suppressed the proliferation of stromal cells and inhibited hair shaft elongation in cultured hair follicles. Shin [15] reported that DHT induced the expression of androgen receptors and reduced the size of dermal papilla cell spheroids in vitro, mimicking the hair follicle miniaturization in AGA. It was also demonstrated that DHT regulated DPCs’ function and inhibited hair growth. In this study, we found that CSCE restored DHT-induced cell damage in a dose-dependent manner (Figure 2b) and effectively reduced DHT-driven increase in IL-6 and IL-1α productions. (Figure 3c,d).

The Bcl-2 family proteins play a vital role in apoptosis by regulating the permeability of the inner and outer mitochondrial membranes. These proteins interact with each other and activate a series of downstream genes. p53 is involved in the apoptosis of hair follicles during anagenesis [36]. We found that CSCE upregulated bcl-2 expression and downregulated bax and p53 expressions in DHT-incubated cells (Figure 4). It was speculated that CSCE might prolong the anagen phase and delay catagen progression by eliminating DHT induced cell damages.

AGA was associated with the upregulation of SRD5A2 and AR expressions in DPCs [31]. The expression levels of AR and SRD5A2 were found to be higher in the dermal papilla cells of AGA patients than that in the control group [37,38]. Studies have shown that there was a correlation between hair density and the positive expression rate of AR. Cells with high AR expression levels were more sensitive to androgens. CSCE exhibited anti-androgenic activity by reversing the expressions of SRD5A2 and AR in DPCs driven by DHT (Figure 5).

Any damages imposed on DPCs may cause DPCs senescence and lead to hair loss, which disrupts hair follicle homeostasis, loses the ability to induce hair follicle regeneration and inhibits hair follicle differentiation and stem cell growth [39]. The DPCs of male patients with AGA showed premature senescence [9,10,11], which was related to SRD5A2. 11 DHT induced the senescence of DPCs (Figure 6a–c) by causing DNA damage. We observed that CSCE inhibited the enzymatic activity of β-galactosidase (Figure 6d–f), and delayed cell senescence stimulated by DHT. Consistently, p53 has been shown to be a central mediator of cellular senescence by inducing cell cycle arrest [40,41].

In this study, we found that CSCE may support hair growth by abrogating the effects of DHT in cultured DPCs. CSCE restored DHT-induced cell damage in a dose-dependent manner, and effectively reduced the productions of IL-6 and IL-1α driven by DHT incubation. CSCE had a certain anti-apoptotic effect, which increased the expression of bcl-2 and decreased the expressions of bax and p53 in DHT-incubated cells. CSCE also showed an anti-androgenic effect reversing the increase in AR and SRD5A2 expressions in DPCs driven by DHT incubation. In addition, CSCE inhibited β-galactosidase enzyme activity and slowed down cell senescence. Furthermore, the preventive and therapeutic effects of CSCE on AGA and its mechanism still need further evaluation.

## 4. Materials and Methods

### 4.1. Preparation of Camellia Seed Cake Extract

The defatted Camellia seed cake (provided by Guangzhou Adolf Innovation Laboratory) was dried and then milled into a coarse powder. Dry powder samples were soaked in 60% ethanol (liquid-solid ratio: 30 mL/g) with ultrasound for an hour at 50 °C. Then, calcium oxide was added and mixed for 3 h to precipitate the saponin components, then centrifuged at 4000 r/min for 10 min at 20 °C and the supernatant was filtered and freeze-dried to obtain Camellia seed cake extract (abbreviated as CSCE). The supernatant containing CSCE was freeze-dried.

### 4.2. Phytochemical Analysis by UPLC-MS

Characterization of CSCE was acquired using UPLC Q-TOF-MS (Waters Corporation, Millford, MA, USA). Lipids were separated using an Acquity UPLC BEH C18 column (2.1 × 50 mm, 1.7 μm, Waters), and eluted by an analyte-specific gradient of 0.1% formic acid solution and acetonitrile. The column was maintained at 45 °C and 5 μL was injected with a 0.3 mL/min flow rate. Electrospray ion source (ESI) was working under positive and negative ion electrospray ionization mode. The monitoring parameters were optimized as follows: 30 Volts cone voltage, 3.5 kV capillary voltage; 700 L/h desolvation gas flow rate; 6/20 Volts collision energy; 100 °C source block temperature; 400 °C desolation temperature.

### 4.3. Cell Culture

DPCs were bought from Qingqi Biotechnology Development Co., Ltd. (Shanghai, China), and were cultured in DMEM medium (Gibco, Carlsbad, CA, USA) with FBS (10%, Gibco, Carlsbad, CA, USA), penicillin (100 U/mL) and streptomycin (100 μg/mL, Hyclone, Logan, UT, USA) at 37 °C in a humidified incubator containing 5% CO_2_. Cell lines from passages 3 to 15 were used in the following assays.

### 4.4. Cell Viability Assay

DPCs were grown at a seeding density of 1 × 10^4^/well on a 96-well plate which was incubated at 37 °C for 24 h. Then, the cells were subjected to treatment with different concentrations of CSCE for 24 h. Subsequently, MTT (0.5 mg/mL, 100 μL/well) (Sigma-Aldrich Co., St. Louis, MO, USA) was added and incubated with the cells for 4 h. The medium was discarded, and DMSO (Sigma-Aldrich Co., St. Louis, MO, USA) was used to dissolve the MTT formazan precipitate. Absorbance at 490 nm was measured on a microplate reader. Each assay was repeated three times.

### 4.5. IL-6 and IL-1α ELISA Detection

Cells were grown on a 12-well plate for 24 h at a starting density of 1 × 10^6^/well. Then, the cells were subjected to treatment with different concentrations of CSCE for 24 h. The supernatants were collected to detect the secretion of cellular inflammatory factors using IL-6 and IL-1α ELISA kits (Beyotime Beyotechnology, Co., Shanghai, China).

### 4.6. Senescence-Associated β-Galactosidase (SA-β-Gal) Assay

Cells were incubated with DHT (Sigma-Aldrich, St. Louis, MO, USA) and different concentrations of CSCE for 24 h. After incubation, the culture medium was removed by suction and the cells were washed twice with PBS. Then, the cells were fixed with β-galactosidase staining fixative (125 μL per well, Beyotime Beyotechnology, Co.) at room temperature for 15 min. The cell fixative was removed by aspiration, and the cells were washed 3 times with PBS (Hyclone, USA). Subsequently, the cells were incubated in a staining working solution (125 μL/well, Beyotime Beyotechnology, Co.) overnight at 37 °C. Cell staining was observed and photographed.

### 4.7. Quantitative Measurement of mRNA Expression of Cytokines

Cells were cultured on T25 flasks at a seeding density of 5 × 10^6^/well for 24 h. Then, the cells were incubated with different concentrations of DHT solutions (non-cytotoxic) for another 24 h before getting collected. Total RNA was extracted using the Rneasy^®^ RNA extraction kit (Beyotime Beyotechnology, Co.), and cDNA was synthesized on a ThermoCycler with the cDNA synthesis kit. RT-PCR was performed to determine the transcription levels. Primers used are as follows: GAPDH sense 5′-GGAAGCTTGTCATCAATGGAAATC-3′, antisense 5′-TGATGACCCTTTTGGCTCCC-3′; AR sense 5′-AATCCCACATCCTGCTCAAGAC-3′, antisense 5′-GGAAAGTCCACGCTCACCAT-3′; SRD5A2 sense 5′-TCAATCGAGGGAGGCCTTATC-3′, antisense 5′-CCCAAGCTAAACCGTATGTCTG-3′; Bcl-2 sense 5′-GAGGATTGTGGCCTTCTTTGAG-3′, antisense sense 5′-ACAGTTCCACAAAGGCATCCC-3′; P53 sense 5′-GAGGTTGGCTCTGACTGTACC-3′ antisense sense 5′-TCCGTCCCAGTAGATTACCAC-3′; BAX sense 5′-TTTTGCTTCAGGGTTTCATCCA-3′; antisense sense 5′-TGCCACTCGGAAAAAGACCTC-3′.

### 4.8. Statistics 

Statistical analysis was performed through the GraphPad Prism8 software, and ANOVA was applied for comparison between different groups. The results were presented as the means ± SD, and a significant different was considered to be when *p* < 0.05.

## 5. Conclusions

In summary, this work reported the anti-apoptotic and anti-androgenic activities of the Camellia seed cake extract (CSCE) in cultured human DPCs. CSCE was able to restore DHT-induced cell damage and reduce the production of IL-6 and IL-1α in DHT-treated DPCs. CSCE also increased the expression of bcl-2, decreased the expression of bax and p53, and reversed the increase in AR and SRD5A2 expression in DPCs induced by DHT. Moreover, CSCE could inhibit β-galactosidase enzyme activity and thereby slow down the DHT-induced cellular senescence. Therefore, CSCE may have the potential to prevent and alleviate AGA by abrogating the effect of DHT.

## Figures and Tables

**Figure 1 molecules-27-06443-f001:**
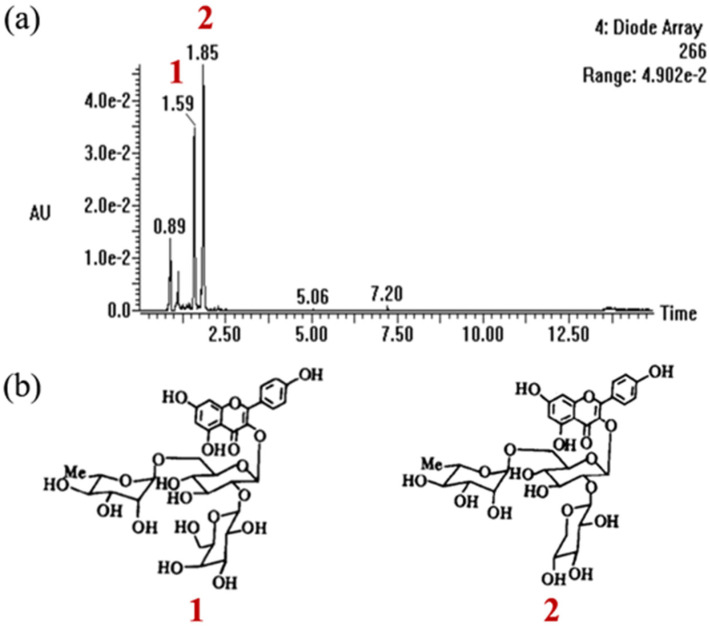
Phytochemical characterization of CSCE (**a**) and chemical structure of components (**b**).

**Figure 2 molecules-27-06443-f002:**
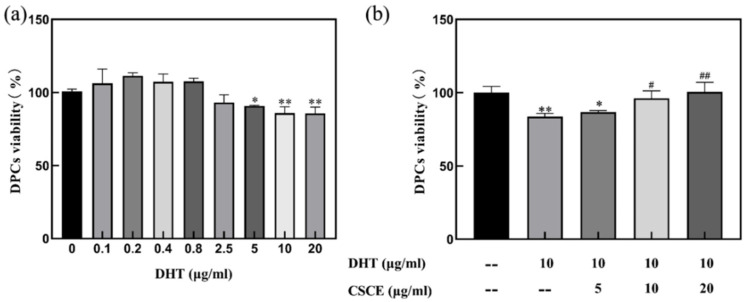
Cell viability of DPCs. (**a**) Viability of DPCs treated with different concentrations of DHT. (**b**) The effect of CSCE on DHT-induced cell damage in DPCs. (Compared with blank control group, ** p* < 0.05, *** p* < 0.01; compared with 10 μg/mL DHT group, *^#^ p* < 0.05, *^##^ p* < 0.01).

**Figure 3 molecules-27-06443-f003:**
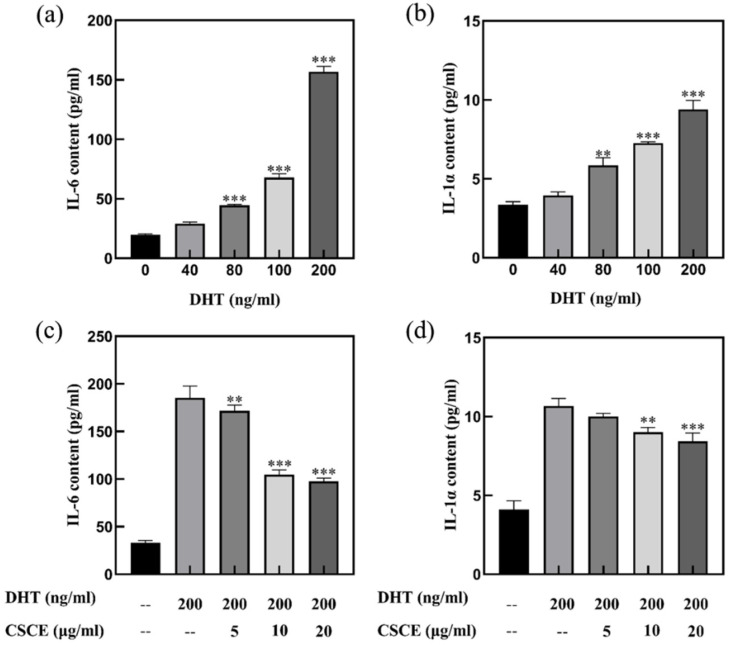
DHT induced the release of IL-6 (**a**) and IL-1α (**b**) inflammatory factors in DPCs, and the induction activity was inhibited by CSCE (**c**,**d**). (Compared with blank control group, *** p* < 0.01, **** p* < 0.001).

**Figure 4 molecules-27-06443-f004:**
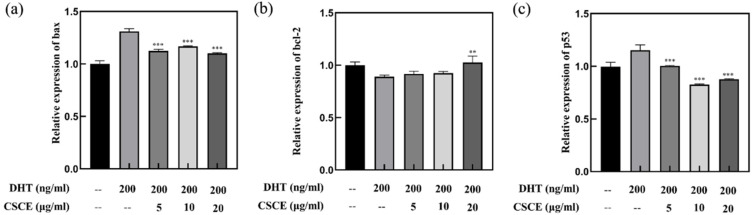
The effect of CSCE on the expression of apoptosis signaling molecules in DPCs: (**a**) bax; (**b**) bcl-2; (**c**) p53. (Compared with DHT model group, *** p* < 0.01, **** p* < 0.001).

**Figure 5 molecules-27-06443-f005:**
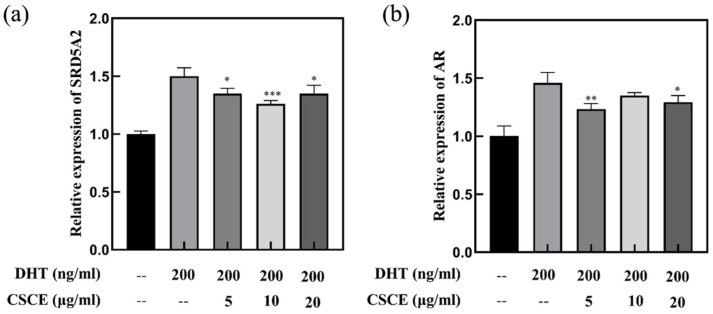
Effects of CSCE on the expressions of SRD5A2 (**a**) and AR (**b**) in DPCs. (Compared with DHT model group, ** p* < 0.05, *** p* < 0.01, **** p* < 0.001).

**Figure 6 molecules-27-06443-f006:**
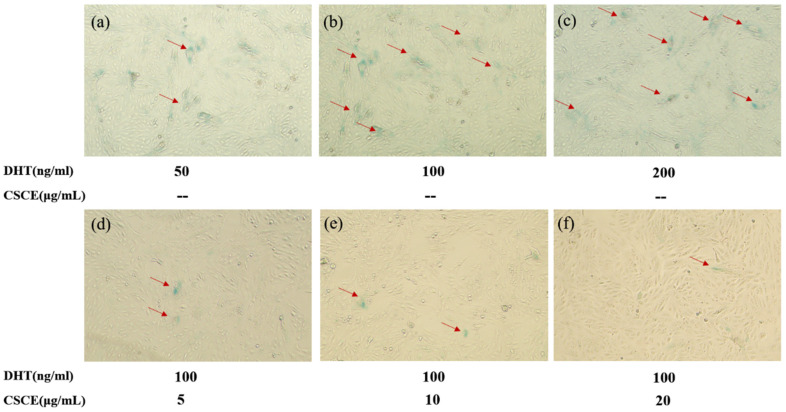
SA-β-Gal in DPCs of different groups was stained and photographed. (**a**–**c**) 50 ng/mL, 100 ng/mL and 200 ng/mL DHT-treated group; (**d**–**f**) 100 ng/mL DHT and 5, 10, and 20 μg/mL CSCE co-treated DPCs group.

## Data Availability

Not applicable.

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
