# Peer review of "Camellia Seed Cake Extract Supports Hair Growth by Abrogating the Effect of Dihydrotestosterone in Cultured Human Dermal Papilla Cells"

_molecules, 2022, doi:10.3390/molecules27196443_

Round 1

Reviewer 1 Report

Dear Authors!

My comments are in attached file

Author Response

Dear reviewers!

Reply please find in attachment.

Best Regards

Authors

Reviewer 2 Report

The authors report a study on the definition of the molecular mechanism regulated by the CSCE in slowing androgynous alopecia. The authors have already reported the anti-alopecia effects of CSCE extracts in theirs previous manuscript in an animal model (mouse). Now, in this manuscript they define the molecular signaling involved. In particular, they demonstrated that CSCE extracts reduce the release of inflammatory cytokines (IL-1a and IL-6) in dermal papilla cells (DPC) after treatment with dihydrotestosterone by ELISA assay. Furthermore, CSCE extracts act on the apoptotic signals of BAX, BCL2 and P53. Finally, the authors measure a reduced expression of the AR androgen receptor mRNA and testosterone reductase after treatment with CSCE extracts. So the authors propose their extract as a potential treatment for androgynous alopecia.

The manuscript is clear and well written. The experimental design and the statistical approach are correct.

Authors should better format bibliographic notes in the text. It was not easy to read the text with the reference numbers without brackets.

Author Response

(The authors gave the same response as above.)

Reviewer 3 Report

INTRODUCTION
Lines 32-39 Cite appropriate references.
Lines 40-43 Cite appropriate references.
Lines 79-85 Please explain in detail why CSCE was extracted with 60% ethanol, adding data from other concentrations of ethanol.

Line 97 Is the DPC used from balding or normal? Indicate gender, age, and number of passages.

Line 144-161 You mentioned main two flavonoids, please show the HPLC data and NMR data.

Lines 170-178 Please show the data of CSCE cytotoxicity when not induced by DHT.

Lines185-197 Show the effect of CSCE on IL-6 and IL-1α production in the absence of DHT induction.

Lines203-220 Show the effect of CSCE on bax, bcl2 and p53 in the absence of DHT induction.

Lines234-245 Show the effect of CSCE on β-galactosidase-positive DPCs in the absence of DHT induction.

Discussion and Conclusion
The above data have been added and will be peer-reviewed.

Author Response

(The authors gave the same response as above.)

Round 2

Reviewer 3 Report

Responses to the comments are not properly reflected in the revised version.
